# Infectious disease surveillance in U.S. jails: Findings from a national survey

**Morgan Maner** [1]*, **Marisa Omori** [2], **Lauren Brinkley-Rubinstein** [1], **Curt G. Beckwith** [3], **Kathryn Nowotny** [4]

**1** Department of Social Medicine, Center for Health Equity Research, University of North Carolina at Chapel Hill, Chapel Hill, North Carolina, United States of America, **2** Department of Criminology and Criminal Justice, University of Missouri, Saint Louis, MO, United States of America, **3** Division of Infectious Diseases, Alpert Medical School of Brown University, The Miriam Hospital, Providence, Rhode Island, United States of America, **4** Department of Sociology, University of Miami, Coral Gables, FL, United States of America

* morgan_maner@alumni.brown.edu

**Data Availability Statement:** Should there be any requests for our data, the appropriate non-author contact is: University of Miami Human Subject Research Office hsro@miami.edu Kathryn Nowotny (co-author) should also be contacted for

## Abstract

While infectious diseases (ID) are a well-documented public health issue in carceral settings, research on ID screening and treatment in jails is lacking. A survey was sent to 1,126 jails in the United States to identify the prevalence of health screenings at intake and characteristics of care for ID; 371 surveys were completed correctly and analyzed. Despite conflicting Centers for Disease Control (CDC) guidance, only seven percent of surveyed jails test individuals for HIV at admission. In 46% of jails, non-healthcare personnel perform ID screenings. Jails in less urban areas were more likely to report healthcare screenings performed by correctional officers. Survey findings indicate that HIV, HCV and TB testing during jail admissions and access to PrEP are severely lacking in less urban jails in particular. Recommendations are provided to improve ID surveillance and address the burden of ID in correctional facilities.

## Introduction

Populations that end up in prisons and jails have a significantly higher burden of ID compared to the general population [1], and prisons are not designed to mitigate ID spread. Overcrowding, poor ventilation, and delays in access to medical care are several additional factors that contribute to risk of ID in carceral settings [2–5]. Overcrowding is a result of mass incarceration: the United States has an incarceration rate of 698 per 100,000 people [6]. As of 2011–2012, 14% of people in jails had ever had an infectious disease, including tuberculosis (TB), hepatitis C virus (HCV), or another sexually transmitted disease [1]. People in prisons and jails are at typically at high risk for ID before incarceration and once inside a carceral facility, the risk only increases; human-immunodeficiency virus (HIV) prevalence in carceral facilities is 3–5 times higher than in the general population [7] and the seroprevalence of HCV in carceral populations was estimated to be 18% in 2015 [8].

Despite Centers for Disease Control (CDC) guidance that calls for universal opt-out HIV testing in prisons, where all incarcerated people would be tested for HIV unless explicitly refused [9] only 18 prison systems in the U.S. conducted HIV testing between 2010 and 2016

data access. Her email address is: kathryn.
nowotny@miami.edu.

**Funding:** This research was supported by the
Criminal Justice Research Training (CJRT)
Program (R25DA037190) and Miami Center for
AIDS Research (P30AI073961). The funders had
no role in study design, data collection and
analysis, decision to publish, or preparation of the
manuscript.

**Competing interests:** The authors have declared
that no competing interests exist.

[8]. In a recent survey of health screening and routine care practices in 2014, in which medical directors of forty of the largest jails were surveyed on HCV testing and treatment practices, only one jail of twenty-three that responded to the survey provided routine HCV testing, and only 35 percent of surveyed jails followed CDC recommendations in providing opt-out HIV testing at admission [10].

Jails house people with a high prevalence of ID that serve short-term sentences after being accused or convicted of a crime [1]. The criminalization of drug use has generated a cycle of repetitive incarceration of many individuals infected with HIV, hepatitis B virus (HBV), hepatitis C virus (HCV) as a result of high-risk drug use [11]. In 2019, the weekly turnover rate in jails was 53%, and the average length of stay across the nation was 26 days [12]. 82% of individuals released from state prisons in 24 states were arrested at least once in the 10 years following their release in 2008 [13]. Jails in less urban areas might be especially susceptible to ID outbreaks and other health issues because pretrial incarceration rates have grown the most in rural counties in the past several years [14,15] yet may lack the same health infrastructure.

Individuals recently released from jails and prisons are at a particularly heightened risk for HIV acquisition due to substance use, risky sexual behavior, lack of healthcare access, and homelessness [2]. The same holds true for HCV [16]. Opioid-related overdoses are a leading cause of death in individuals re-entering their communities after incarceration [17]. Opioid substitution therapy, a highly successful treatment for OUD, is only available in a small number of prisons in the United States (specifically, Riker's Island in New York, and a selection of prisons in Baltimore, Philadelphia, and Rhode Island), and no prisons provide needle and syringe exchange programs [18].

Education and access to treatments to prevent HIV, such as pre-exposure prophylaxis (PrEP) among high-risk HIV-negative persons is particularly limited in jail settings and upon release. PrEP is highly effective at preventing HIV acquisition when taken as prescribed; it reduces the risk of acquiring HIV from sexual behaviors by 99% and from intravenous drug use by 74% [19]. In a recent study of one unified jail system, 88% of surveyed incarcerated men who have sex with men (MSM) had never even heard of PrEP. Despite this, study participants were very interested in using PrEP as a preventative measure [20].

Public health surveillance and treatment in carceral settings are vital opportunities to protect the health of people inside prisons and jails in addition to the surrounding community [21]. Research suggests that these interventions are also achievable; a study in Rhode Island found that rapid HIV testing, along with risk reduction counseling, was feasible and highly acceptable among people held in jail [22]. Many public health advocates, physicians and researchers have created guidelines and cascades to make ID surveillance, prevention, and treatment possible in carceral settings [23–27].

To provide a comprehensive update on current practices around ID and health surveillance in U.S. jails, we administered a survey to all jails with an American Jail Association (AJA) membership. The survey assessed ID surveillance procedures at intake and the provision of PrEP in addition to non-urgent care, staffing and substance use treatment.

## Materials and methods

A survey modeled from the 2011 BJS National Survey of Prison Health Care, [28] which identified health screening practices related to ID, cardiovascular disease, and mental health, was developed and administered to jails across the United States. The survey was adapted for the jail setting by an Expert Advisory Group composed of jail administrators and AJA representatives, along with a physician with experience delivering healthcare in jails.

There were 1,126 jails that met the following inclusion criteria: (1) were members of the AJA; (2) had a valid phone number and/or email address available for the jail's primary contact listed with the AJA; (3) were included in the 2013 Bureau of Justice Statistics (BJS) Census of Jails; and (4) were not located in states with combined jail/prison systems (AK, CT, DE, HI, RI, and VT). Jails operated by the federal government or Bureau of Indian Affairs were excluded so that only county and regionally operated jails were included. The first wave of surveys was administered during the last week of June 2018; data collection occurred during a six-month period. Consent was not obtained from research participants as the survey is focused only on the policies and procedures of organizations.

The survey included four domains: medical intake, non-urgent care, medical services and staffing, and substance use treatment. A miscellaneous section of the survey asked participants about PrEP referral, accreditation, and the use of telemedicine. Respondents were also asked to voluntarily submit copies of their medical intake forms along with the completed survey. The following definition of medical intake was provided for participants:

> "For the purposes of these questions, medical intakes are the screening questions and activities conducted by health or custody staff after admission, before assigning someone to a housing unit. Medical intake does NOT include screening you do at the front door, while the arresting officer is still present, to decide whether you will accept or divert the arrestee; a more in-depth evaluation that is done by a nurse or practitioner, typically called the 14-day health assessment."

Respondents were asked to indicate the nature of both screening and testing practices for viral hepatitis, gonorrhea, chlamydia, tuberculosis (TB), and HIV/AIDS. For the purposes of this survey, screening referred to asking incarcerated individuals to self-report their ID status. The response categories were coded as the following: (1) screen and test all incarcerated people at admissions; (2) screen all incarcerated people by asking them to self-report their status and offer testing for at least some incarcerated people; (3) screen all incarcerated people by asking them to self-report their status but do not offer any testing; (4) screen only some incarcerated people by asking them to self-report their status and do not offer any testing; (5) do not screen or test at all (not provided); and (6) don't know. For answers that indicated some incarcerated people, a free-response question was provided to explain further. For the full survey, see S1 File.

Primary contacts were identified at each jail. For jails with multiple individual AJA memberships, the highest-ranking member, or the member with the most relevant title (e.g., Jail Administrator, Sheriff) was selected as the primary contact. For jails that only had one member and no agency membership, that person was designated as the primary contact regardless of their title. If no email address was listed for the primary contact, the county/jail website was searched for alternative contact information.

Primary contacts with an email address were sent the survey through email and were offered to complete the survey using Qualtrics or to download a PDF version of the survey to return by postal mail/scanned email. Primary contacts without an email address were sent a survey by postal mail with a self-addressed return stamped envelope. When surveys were returned due to incorrect contact information, attempts were made to find updated contact information. Non-responders were contacted by telephone to see if they received the survey and to encourage participation. Additional surveys were sent as requested. A second wave of surveys were administered to non-responders with the last surveys being sent during January 2019.

The AJA membership records were cleaned in Stata 15, and the addresses were geocoded to the county using ArcGIS 10.5.1. The geocoding process entails transforming the physical address into x/y coordinates, and then spatially joining the coordinates to county boundaries. 2010 Census boundaries were used. The geocoded AJA jails were then merged with the BJS jails using the county-level FIPS code and a jurisdiction crosswalk file used by the BJS. BJS uses 6 categories for jurisdiction size based on the number of beds available at the jail: less than 50, 50–99, 100–249, 250–499, 500–999, and 1,000 or more. The survey data was coded using R and analyzed with SAS Studio. Probability weights were calculated based on BJS designations of jurisdiction size (see S1 Table).

## Results

Out of the 1,126 jails sampled, 376 (33.4%) returned surveys, 81(7.19%) declined to participate, and 674 (59.9%) did not respond. Sixty-four jails (17.0%) also submitted copies of their medical intake forms. Five surveys were discarded because they were incorrectly completed, for a final sample size of 371. The minimum response rate ranges from 28.8% among jails with fewer than 50 incarcerated people to 45.9% among jails with 1,000 or more incarcerated people. The adjusted response rate ranges from 34.4% to 51.7%, respectively. Our sample represents 15.1% of all jails but includes the five largest jails based on reported 2016 year-end total population [29].

Overall, most jails do not offer HCV, HIV or TB testing at medical intake (Table 1). 115 jails provided some qualifying information on when individuals were tested for ID. 30 jails specifically mentioned that they performed testing if the incarcerated person self-reported a

**Table 1.  HIV, TB, and HCV surveillance procedures at intake.**

|  | Number of jails (n = 371) | Percent of jails |
| --- | --- | --- |
| **HCV** |  |  |
| All screened with testing | 20 | 5.26 |
| All screened with some testing | 30 | 8.03 |
| All screened with no testing | 250 | 67.3 |
| Some screened with no testing | 29 | 7.76 |
| No screening or testing | 43 | 11.6 |
| Don't know | 0 | 0 |
| **HIV** |  |  |
| All screened with testing | 25 | 6.65 |
| All screened with some testing | 44 | 11.9 |
| All screened with no testing | 235 | 63.4 |
| Some screened with no testing | 25 | 6.65 |
| No screening or testing | 40 | 10.8 |
| Don't know | 2 | 0.55 |
| **TB** |  |  |
| All screened with testing | 118 | 31.9 |
| All screened with some testing | 31 | 8.31 |
| All screened with no testing | 176 | 47.4 |
| Some screened with no testing | 20 | 5.26 |
| No screening or testing | 27 | 7.20 |
| Don't know | 0 | 0 |

[a]Screening refers to asking incarcerated individuals to self-report their ID status.

positive status and 26 jails stated that they performed tests upon request of the incarcerated person. 26 jails stated that they performed tests if the incarcerated person disclosed symptoms or reported specific risks. Five jails specifically mentioned opt-in testing and six jails specifically mentioned opt-out testing (these were not mutually exclusive). Although definitions were not provided to respondents, opt-in testing generally means that the test is "presented so the patient would be expected to understand the default is to not test unless he or she states agreement," [30]. Alternatively, an opt-out testing approach requires informing patients (e.g., through a patient brochure, practice literature/form, or discussion) that a test will be performed and that they may decline or defer the test [31].

Larger jails were more likely to offer at least some testing for infectious diseases compared to smaller jails. For example, half of jails with a 1,000+ inmate jurisdiction size (54.4%) screened through self-report and tested all incarcerated people for TB during medical intake. The rates for other sizes in descending order were as follows: 46.0% for 500–999 jail resident jurisdiction size, 40.8% for 250–499 jurisdiction size, 27.7% for 100–249 jurisdiction size, 27.6% for 50–99 jurisdiction size, and 12.9% for jurisdiction size of fewer than 50 incarcerated people.

## HCV

Approximately two-thirds of surveyed jails report HCV screening only for HCV surveillance at intake, and only five percent of surveyed jails report offering HCV testing for all individuals at intake (Table 1). A similar trend follows in rural jails, small/midsize region jails and urban jails; in rural jails, 70% rely on screening only and five percent of rural jails offer HCV testing at intake for all incarcerated individuals (Fig 1). Approximately 70% of jails in small/mid-size regions, 70% of suburban jails and 22% of urban jails rely on screening only to detect HCV at medical intake.

Trends in ID surveillance strategies were more evenly spread out across jails that report screening all individuals for HCV and only testing some depending on certain criteria: 21% of these jails are in rural regions, 31% in small/mid-size, 27% in suburban, 21% in urban regions.

## HIV

Approximately two-thirds of surveyed jails relied on screening for HIV surveillance. Seven percent of surveyed jails offered HIV testing for all individuals at intake (Table 1). 67% of rural jails screened individuals for HIV with no testing offered, whereas only 18% of urban jails screened without offering HIV testing (Fig 2). Screening and testing practices for HIV were similar between jails in rural, small/midsize areas and suburban areas. Surveillance procedures differed the most between suburban and urban jails.

Nine percent of surveyed jails administer PrEP to incarcerated people who request it, and 16% provide a referral to PrEP during discharge planning. PrEP referrals to a community-based healthcare provider at discharge also varied by geographic location; seven percent of rural jails, 14% of small-mid size area jails, 35% of suburban jails and 50% of urban jails provided referrals for PrEP.

## TB

Approximately half of surveyed jails reported relying on screening for TB surveillance at intake, however jails were more likely to test for TB overall than for HIV or HCV (Table 1). 32% of surveyed jails offer TB testing to all individuals at intake; in comparison, approximately seven percent and five percent of all surveyed jails test for HIV and HCV at intake, respectively. Across rural, small/mid-size, suburban and urban jails, differences in screening and

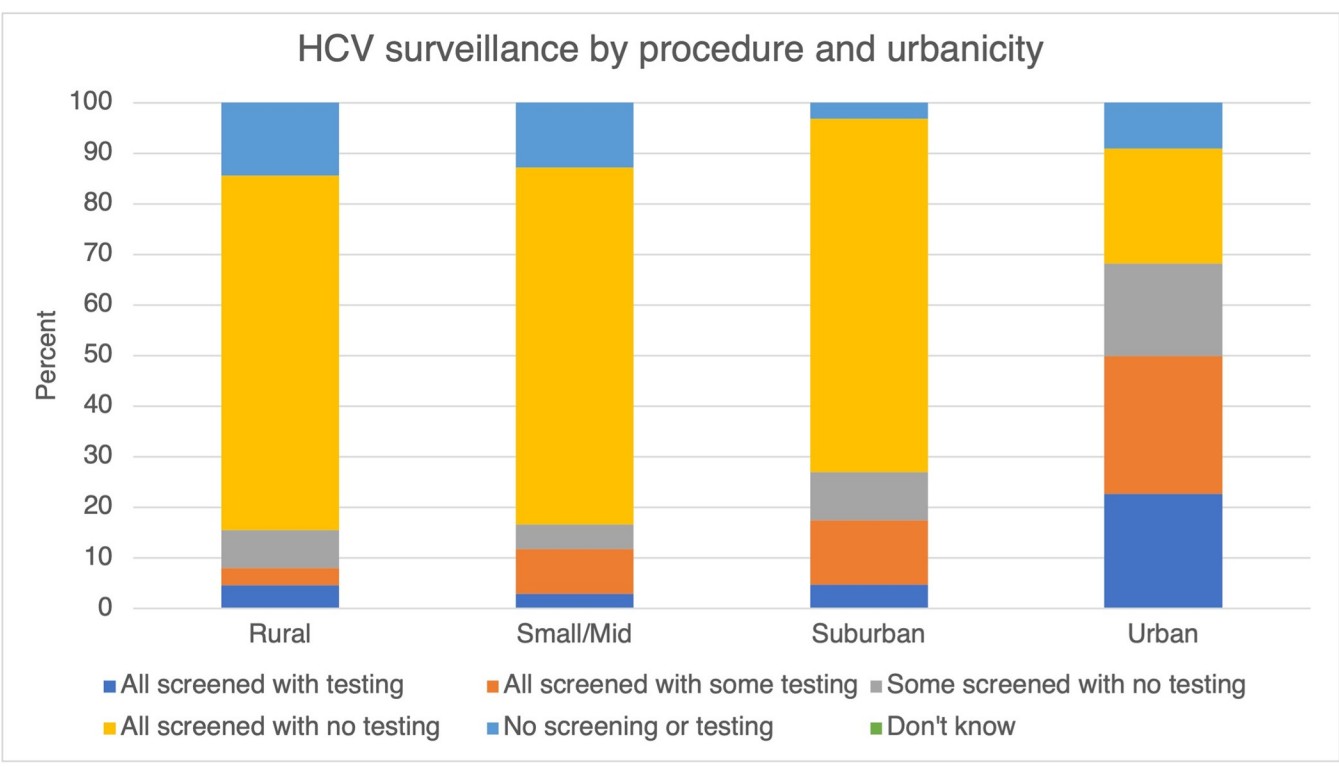

**Fig 1. HCV surveillance by procedure and urbanicity.**

testing practices were the most significant for TB surveillance (Fig 3). All jails in suburban jails offer at least some testing or screening for TB at medical intake.

## Staffing for medical intake

Almost half of surveyed jails reported that correctional officers performed medical intake (Table 2). Staff designations for medical intake varied widely by urbanicity of the jail. Notably, in 66% of rural jails, correctional officers performed medical intake, whereas correctional officers performed medical intake in only 9.5% of urban jails (Fig 4). A registered nurse performed medical intake in 14% of rural jails surveyed and 66% of urban jails.

Staffing for medical intake also differed by jail jurisdiction size; larger jails are more likely to have healthcare staff perform medical intake at admission. Approximately 60% of jails with 50–99 and 100–249 incarcerated people rely on correctional officers to perform medical intake. In 60% of jails with 1000 or more incarcerated people, a registered nurse performs medical intake. A licensed practical or vocational nurse was most likely to perform medical intake in jails with 500–999 incarcerated people.

## Discussion

The response rates for this study were low but adequate when compared to other published rates for voluntary surveys administered by mail, email, and or phone [32,33]. Additionally, non-response bias could have impacted our sample; jails with less robust healthcare services could have opted not to respond to the survey. Additionally, almost half (48%) of surveyed jails were in rural regions, 28% in small/midsize regions, 17% in suburban regions and six percent in urban regions; urban jails are underrepresented in this sample, however, this sample

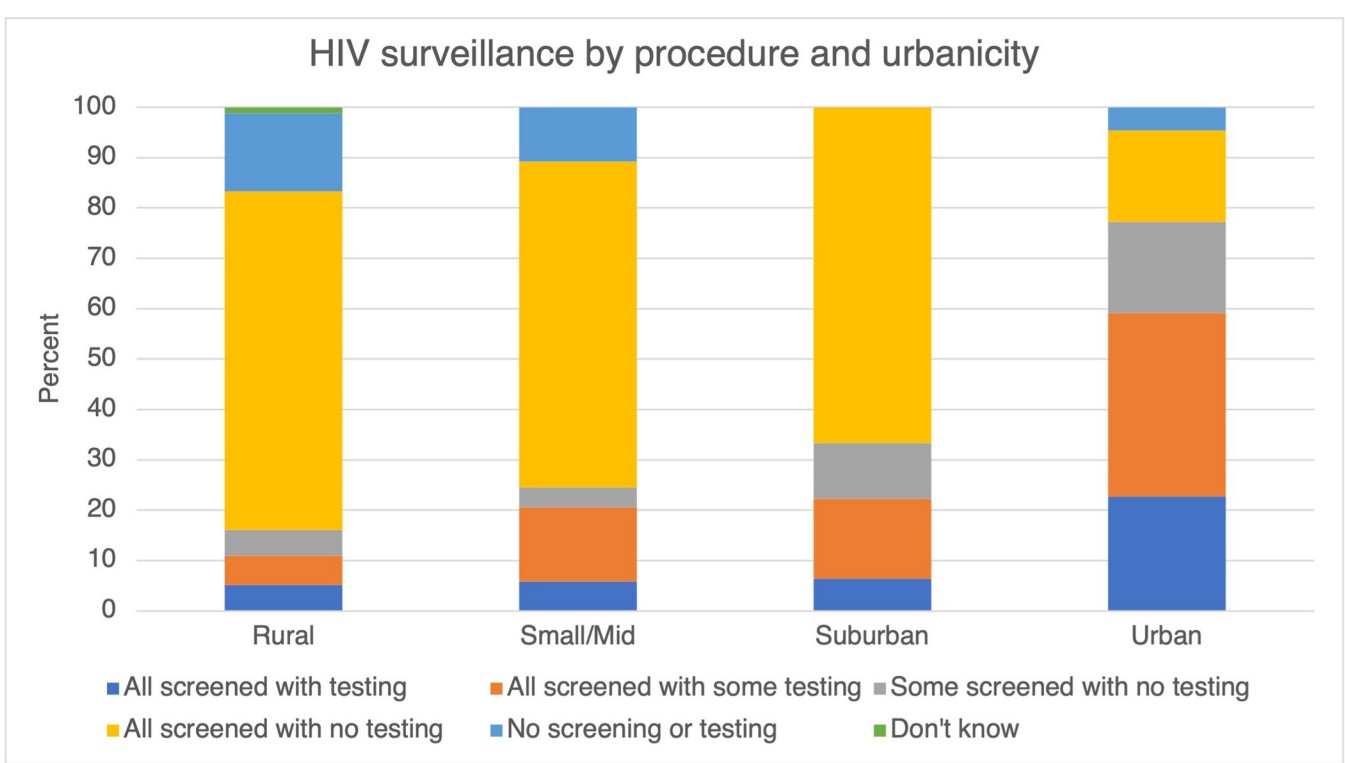

**Fig 2. HIV surveillance by procedure and urbanicity.**

does not significantly diverge from regional trends in incarceration in the United States. As of 2017, incarceration rates are highest in rural regions (265 per 100,000) followed by medium and small metro areas (257 per 100,000), urban areas (200 per 100,000) and suburban areas (178 per 100,000) [15]. Respondents to this survey varied depending on the contact information available for sampled jails. Primary contacts identified for the completion of this survey that were not medical staff might not have had the most realistic or accurate insight into healthcare screening and practices offered at the jail.

Surveyed jails typically did not follow published guidance for ID screening and testing in carceral settings. Approximately two-thirds of surveyed jails relied on self-report to identify HIV and HCV in the incarcerated population, and half of surveyed jails relied on self-report to identify TB status. For HIV in particular, The CDC recommends that universal opt-out testing be offered in all correctional medical clinics, citing that opt-out testing increases HIV diagnoses, reduces stigma associated with HIV testing and preserves staff resources through standard consent and counselling processes [34] which is particularly important in settings where significant staff shortages exist across all sectors of carceral staff [35–37]. For TB surveillance, the CDC recommends that all individuals should be screened by medical staff at entry for TB symptoms at admission [38]. For HCV surveillance, the CDC recommends that all incarcerated people are asked questions regarding risk factors for HCV infection during admission; all people reporting risk factors for HCV infection should be promptly tested for HCV [39].

While there is clear guidance for optimal screening and testing practices in prisons, the AJA does not publish any specific recommendations for jails. Testing rates for surveyed jails were very low in this sample compared to similar surveys of testing practices in prisons and jails [9,28]. People cycle in and out of jails at a much higher rate than in prisons; jails may not be equipped to provide sustainable medical care. Low testing rates from this sample could be

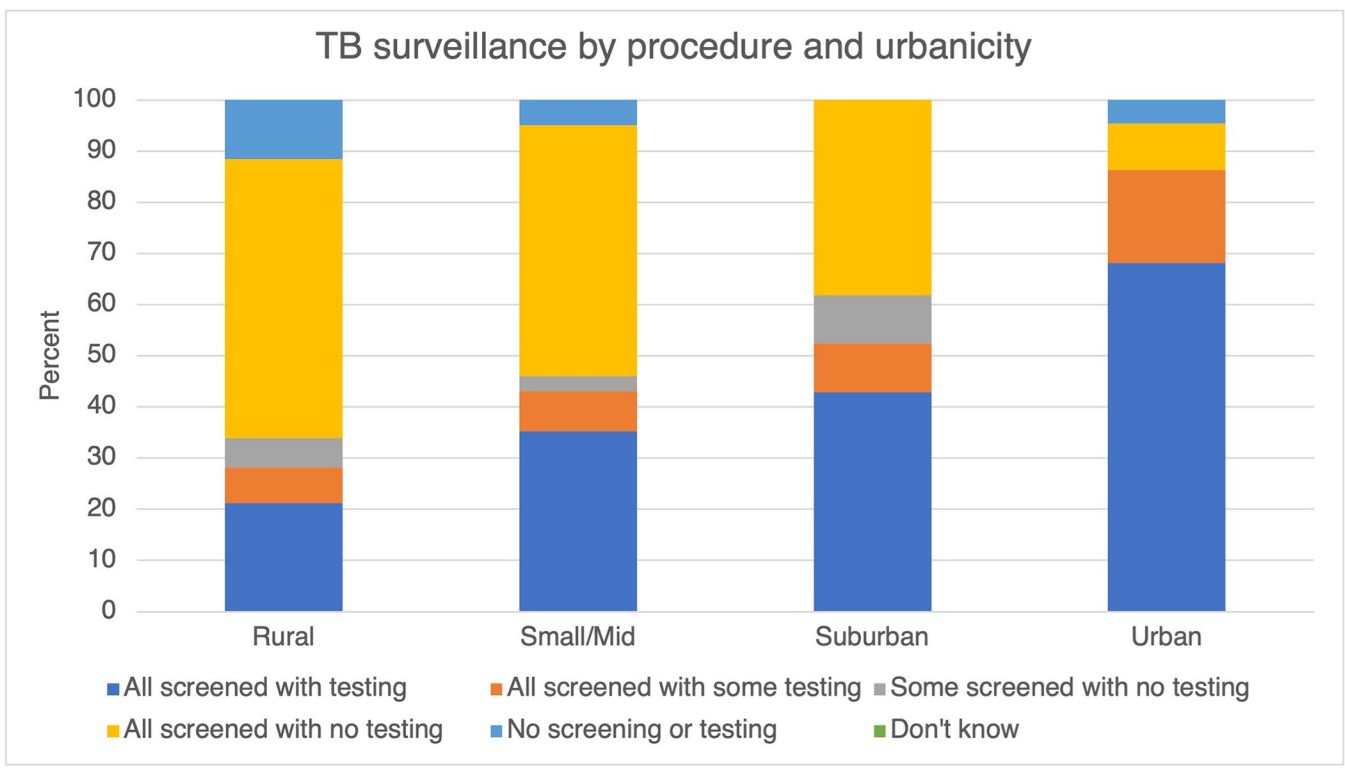

**Fig 3. TB surveillance by procedure and urbanicity.**

due to this lack of infrastructure. Robust ID screening and testing programs require resources and sustained commitment from jail administration and medical staff already challenged by staffing shortages and high staff turnover [35–37].

ID surveillance in jails is a vital opportunity to protect individual and population health both within jails and in the local community upon release. Identifying ID in carceral settings presents an opportunity to also address OUD and SUD as they are commonly syndemic in populations in jails and prisons [40]. Comprehensive healthcare screenings in jails can contribute to a reduction in OUD/SUD-related morbidity and mortality, which occurs at much higher rates in populations that end up in prisons and jails than in the general population [41].

**Table 2. Staffing for medical intake.**

|  | Number of jails | Percent of jails |
|---|---|---|
| **Who performs medial intake at admission?** | | |
| Correctional officer | 172 | 46 |
| Registered nurse | 97 | 26 |
| Licensed nurse | 79 | 21 |
| 'Other medical personnel' | 16 | 4 |
| Medical practitioner | 4 | 1 |
| Civilian staff person | 3 | 1 |

[a] 'Licensed nurse is defined as a licensed practical nurse (LPN) or licensed vocational nurse (LVN). 'Other medical personnel' includes paramedics, EMTs and nurses' aides. 'Medical practitioner' includes physicians [MD/DO], physician assistants and nurse practitioners.

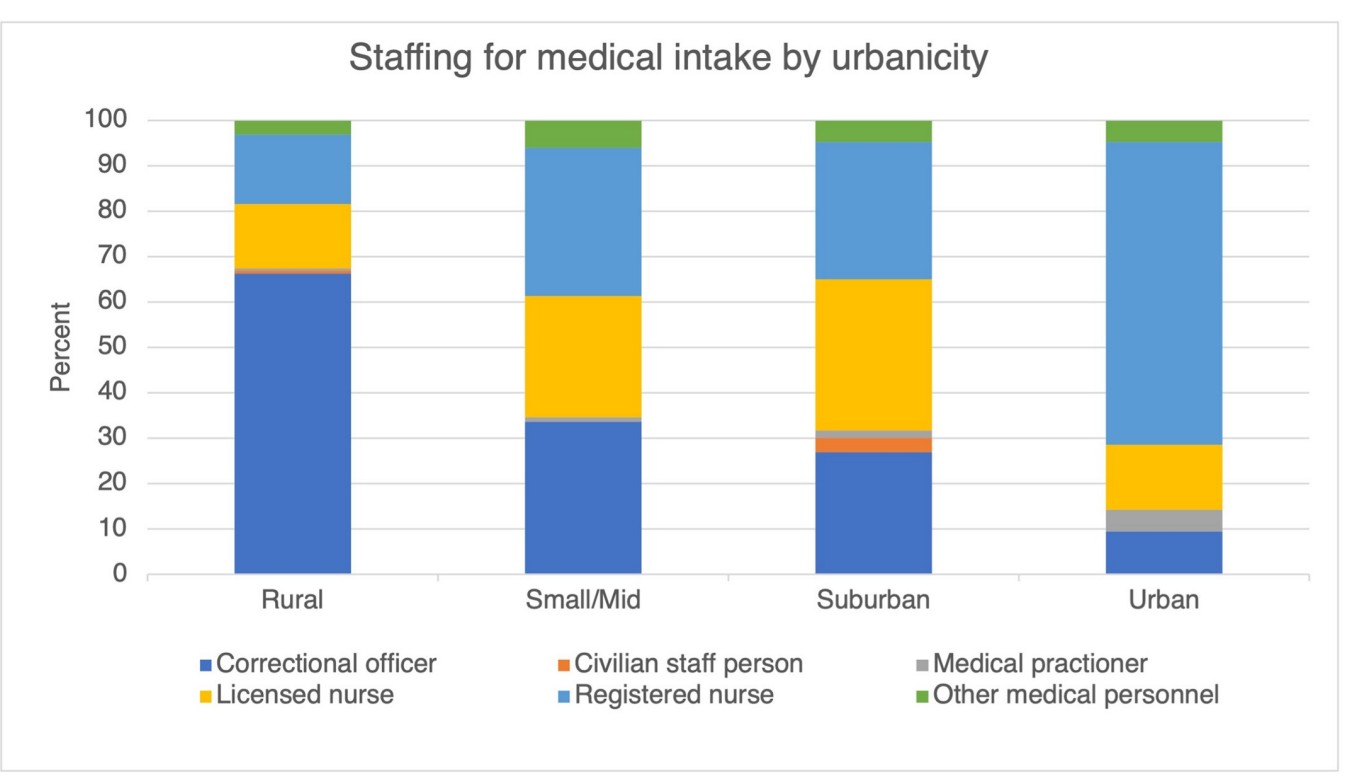

**Fig 4. Staffing for medical intake by urbanicity.**

The expansion of testing programs is needed in a coordinated public health effort that involves local departments of health and community providers for chronic infections such as HIV and HCV.

Typically, individuals return to their communities after jail incarceration without any linkage to community healthcare providers. Only 16% of surveyed jails provided individuals with a referral to a community PrEP provider upon release, and only nine percent provided PrEP inside jail. While this is not surprising, [42] there is significant potential to reduce carceral, community and global HIV burden by starting individuals determined to be at high risk for HIV on PrEP, and by providing linkage to HIV care upon release [43]. Linking jail residents to healthcare at release not only reduces ID burden for carceral populations, but also improves the surrounding population's health [44].

Stark disparities in testing for ID were apparent along the continuum of rural to urban jails. Rural jails in particular face lots of challenges in preventing and mitigating ID burden, most notably due to a lack of healthcare providers in the area. Rural communities as a whole face high rates of intravenous drug use without a corresponding amount of healthcare providers and resources to address it. From 2007–2015, age-adjusted rates of over-dose-related deaths were highest in rural states [45]. Only three percent of primary care providers, which make up the bulk of healthcare providers in rural areas, can prescribe buprenorphine-naloxone to treat OUD [46]. Health screenings should be expanded in rural jails and specifically include testing for ID and concurrent OUD/SUD at intake with a concurrent expansion of ID and OUD/SUD treatment capacity.

In almost half of surveyed jails (46%), CO's performed medical intake. Healthcare staff dedicated and trained to screen individuals for ID and other conditions should be required to

perform medical intake in correctional facilities nationwide. In the carceral environment, there is already a significant amount of stigma and fear of reprisal. Inherently, people in jails are likely to feel reluctant in sharing any personal information with CO's, particularly as it relates to sensitive topics. Furthermore, medical staff are naturally more equipped to identify relevant aspects of the medical history and risk factors for ID that need to be addressed in a carceral environment.

Following this survey's findings, we suggest the following recommendations to improve ID diagnosis and prevention in jails:

1. **Coordinate ID prevention efforts between the Departments of Health and DOCs**
   Health efforts to significantly expand ID screening and delivery of treatment should include early, sustained collaboration between departments of corrections, departments of public health and community providers for chronic infections such as HIV and HCV. These collaborations are vital to providing linkage to care post-release.

2. **Incentivize healthcare providers to work in rural jails**
   Incarceration rates are highest in rural jails despite an overall lack of healthcare providers in rural areas. Healthcare providers must be incentivized to work in rural regions to protect the health of those living inside carceral facilities and the surrounding community.

3. **Limit all medical intake privileges to healthcare staff**
   Correctional staff should not perform medical intake or any screenings for ID, particularly when related sensitive topics such as sexuality and drug use can arise.

4. **Decarcerate individuals to decrease the likelihood and burden of ID**
   Jails are not equipped to manage the burden of ID in overcrowded vulnerable populations nor do they promote the overall health and wellbeing of incarcerated people. A more impactful health intervention will implement alternative rehabilitation strategies that decrease structural inequities and invest in community health strategies, ultimately decreasing the need for prisons and jails.

## Conclusion

Health screenings are an important tool to identify ID and jails provide an important public health opportunity to conduct ID surveillance. Identifying ID protects the health of incarcerated individuals and staff who work in jail settings and is an opportunity to provide linkage to care upon release. Very few jails referred individuals for PrEP despite the high prevalence of HIV in carceral settings. Limitations in staffing must also be addressed, particularly in rural settings where a significantly high proportion of jails rely on correctional officers to perform ID surveillance. Future research should expand published guidelines to identify best practices for health screening and ID surveillance specific to jail settings.

## Supporting information

**S1 Table. Prevalence of jurisdiction size.**
(DOCX)

**S1 File. National Survey of Health Care in U.S. Jails.**
(PDF)

## Author Contributions

**Conceptualization:** Morgan Maner, Kathryn Nowotny.

**Data curation:** Marisa Omori.

**Formal analysis:** Morgan Maner.

**Funding acquisition:** Curt G. Beckwith, Kathryn Nowotny.

**Methodology:** Curt G. Beckwith, Kathryn Nowotny.

**Project administration:** Kathryn Nowotny.

**Supervision:** Lauren Brinkley-Rubinstein.

**Writing – original draft:** Morgan Maner.

**Writing – review & editing:** Morgan Maner, Marisa Omori, Lauren Brinkley-Rubinstein, Curt G. Beckwith, Kathryn Nowotny.

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
