## [Decision Letter · Decision Letter 0]

7 Jun 2022

PONE-D-22-05652Infectious disease surveillance in U.S. jails: findings from a national surveyPLOS ONE

Dear Dr. Maner,

Thank you for submitting your manuscript to PLOS ONE. After careful consideration, we feel that it has merit but does not fully meet PLOS ONE’s publication criteria as it currently stands. Therefore, we invite you to submit a revised version of the manuscript that addresses the points raised during the review process.

We look forward to receiving your revised manuscript.

Kind regards,

Sungwoo Lim, DrPH

Academic Editor

PLOS ONE

Journal Requirements:

2. You indicated that ethical approval was not necessary for your study. We understand that the framework for ethical oversight requirements for studies of this type may differ depending on the setting and we would appreciate some further clarification regarding your research. Could you please provide further details on why your study is exempt from the need for approval and confirmation from your institutional review board or research ethics committee (e.g., in the form of a letter or email correspondence) that ethics review was not necessary for this study? Please include a copy of the correspondence as an ""Other"" file.

Reviewers' comments:

Reviewer's Responses to Questions

**Comments to the Author**

1. Is the manuscript technically sound, and do the data support the conclusions?

Reviewer #1: Yes

2. Has the statistical analysis been performed appropriately and rigorously? 

Reviewer #1: Yes

3. Have the authors made all data underlying the findings in their manuscript fully available?

Reviewer #1: Yes

4. Is the manuscript presented in an intelligible fashion and written in standard English?

Reviewer #1: Yes

5. Review Comments to the Author

Reviewer #1: This is a very important contribution that provides evidence to back up what scholars and public health practitioners have long suspected to be the case but have had little data to show. I strongly recommend publication, as I believe this study is well done, original, and has the potential to have high impact in shifting policy and oversight with substantial population-level benefits for public health.

I have a few suggestions for possible revisions with the goal of potentiating this study's potential for practical effect. The authors' should use their judgment in whether these suggestions are in fact practical within the confines of their article.

1) The Introduction opens on line 33 by noting that "Populations that end up in prisons and jails have a significantly higher burden of ID compared to the general population." Given that this article will likely find broad traction in media and with non-specialized audiences, I recommend indicating why this is the case: populations that are subjected to higher rates of policing and arrest are the same populations that have least access to healthcare, especially to primary care and preventative medicine. As a result, they enter carceral settings with a higher rate of both infectious and chronic diseases than the age-adjusted general population. Linking poor healthcare access and quality prior to incarceration to the medical presentations of incarcerated people once inside jails and prisons is important, I believe, because it then allows us to point to multiple points of potential policy action to address the public health factors involved the public health phenomena at issue in this study. This allows public health researchers to emphasize that health and healthcare in communities is in interrelation with health and healthcare inside jails and prisons, which, in turn, leads to greater potential to generate policy action to address root causes of criminalization and incarceration (ie, public abandonment of criminalized communities) rather than simply reactive redress.

2) A very minor rhetorical point: When discussing low rates of PrEP referral and provision on line 268, I recommend reframing this so as not to say "this is not surprising." While I agree with the authors that for specialists in this area who know about the normalization of healthcare neglect inside jails and prisons, this is not surprising, rhetorically, this detracts from their point. This should be surprising! And making the fact of the abject healthcare neglect and public health failures noted in this article maximally striking to general audiences, as they should be, may be a better framing strategy. This entire article should be shocking from the perspective of public health in one of the wealthiest nations on Earth.

3) While the authors appropriately focus on testing at time of admission (ie, entrance testing), I recommend also discussing testing upon release (ie, exit testing). From a public health perspective, there are two major points of spread to which this study can speak: 1) intra-facility transmission and 2) transmission of facility-acquired infections to broader communities both via staff circulation and then also upon release of incarcerated people back to their home communities. Testing at time of admission addresses point 1. But given that many people leave jails and prisons with new infections that they acquired while inside, standardized testing upon release with appropriate referrals to healthcare on the outside is another important focus for public health policy. This would not only help reduce community spread of infectious disease but would also facilitate and encourage ongoing healthcare connections on the outside to help mitigate the long-term health harms incurred by incarceration.

4) In order to help this study maximally potentiate effective redress of what it documents, I recommend including a brief discussion of the legal aspects of healthcare failures in carceral contexts. These institutions have a constitutional duty to provide proper healthcare to incarcerated people, but often very little will or dedicated resources to do so. There is a perverse incentive, then, not to do health screening tests that incur liability and responsibility for providing proper care. Additionally, healthcare oversight and regulatory systems to ensure that carceral institutions are meeting their legal responsibilities are famously lacking in the US. One body with power to institute sanctions, put facilities under federal control and bypass local laws and obstructions, etc. is the DOJ's Civil Rights Division. I would thus recommend a brief outline of the available mechanisms for redress in order to guide readers towards pushing the DOJ to act to rectify the problems and violations of legal duties that this study highlights.

6. PLOS authors have the option to publish the peer review history of their article (what does this mean?). If published, this will include your full peer review and any attached files.

Reviewer #1: No

---

## [Author Response · Author response to Decision Letter 0]

12 Jul 2022

Reviewer’s comments to author:

1. The Introduction opens on line 33 by noting that "Populations that end up in prisons and jails have a significantly higher burden of ID compared to the general population." Given that this article will likely find broad traction in media and with non-specialized audiences, I recommend indicating why this is the case: populations that are subjected to higher rates of policing and arrest are the same populations that have least access to healthcare, especially to primary care and preventative medicine. As a result, they enter carceral settings with a higher rate of both infectious and chronic diseases than the age-adjusted general population. Linking poor healthcare access and quality prior to incarceration to the medical presentations of incarcerated people once inside jails and prisons is important, I believe, because it then allows us to point to multiple points of potential policy action to address the public health factors involved the public health phenomena at issue in this study. This allows public health researchers to emphasize that health and healthcare in communities is in interrelation with health and healthcare inside jails and prisons, which, in turn, leads to greater potential to generate policy action to address root causes of criminalization and incarceration (ie, public abandonment of criminalized communities) rather than simply reactive redress.

Author response: Thank you for this suggestion. We have included a reference that points to both the disproportionality in policing and poor healthcare access for populations most impacted by infectious diseases prior to incarceration. 

2. A very minor rhetorical point: When discussing low rates of PrEP referral and provision on line 268, I recommend reframing this so as not to say "this is not surprising." While I agree with the authors that for specialists in this area who know about the normalization of healthcare neglect inside jails and prisons, this is not surprising, rhetorically, this detracts from their point. This should be surprising! And making the fact of the abject healthcare neglect and public health failures noted in this article maximally striking to general audiences, as they should be, may be a better framing strategy. This entire article should be shocking from the perspective of public health in one of the wealthiest nations on Earth.

Author response: Thank you for this suggestion. We have made this revision in the manuscript. 

 While the authors appropriately focus on testing at time of admission (ie, entrance testing), I recommend also discussing testing upon release (ie, exit testing). From a public health perspective, there are two major points of spread to which this study can speak: 1) intra-facility transmission and 2) transmission of facility-acquired infections to broader communities both via staff circulation and then also upon release of incarcerated people back to their home communities. Testing at time of admission addresses point 1. But given that many people leave jails and prisons with new infections that they acquired while inside, standardized testing upon release with appropriate referrals to healthcare on the outside is another important focus for public health policy. This would not only help reduce community spread of infectious disease but would also facilitate and encourage ongoing healthcare connections on the outside to help mitigate the long-term health harms incurred by incarceration.

Author response: We appreciate this suggestion, and while testing at release could be a useful public health strategy, there is not sufficient evidence to support this claim. To justify testing upon exit, there needs to be evidence for intra-incarceration transmission and while it may occur, there is not enough data to support this approach. Additionally, the CDC does not currently recommend testing at release in recently published guidance (which has been incorporated in this manuscript). Lastly, as this study is focused on healthcare screenings at intake, a discussion of exit testing is beyond the scope of this paper. 

4) In order to help this study maximally potentiate effective redress of what it documents, I recommend including a brief discussion of the legal aspects of healthcare failures in carceral contexts. These institutions have a constitutional duty to provide proper healthcare to incarcerated people, but often very little will or dedicated resources to do so. There is a perverse incentive, then, not to do health screening tests that incur liability and responsibility for providing proper care. Additionally, healthcare oversight and regulatory systems to ensure that carceral institutions are meeting their legal responsibilities are famously lacking in the US. One body with power to institute sanctions, put facilities under federal control and bypass local laws and obstructions, etc. is the DOJ's Civil Rights Division. I would thus recommend a brief outline of the available mechanisms for redress in order to guide readers towards pushing the DOJ to act to rectify the problems and violations of legal duties that this study highlights.

Author response: Thank you for this suggestion. We have incorporated this revision as an additional recommendation.

---

## [Decision Letter · Decision Letter 1]

19 Jul 2022

Infectious disease surveillance in U.S. jails: findings from a national survey

PONE-D-22-05652R1

Dear Dr. Maner,

We’re pleased to inform you that your manuscript has been judged scientifically suitable for publication and will be formally accepted for publication once it meets all outstanding technical requirements.

Kind regards,

Sungwoo Lim, DrPH

Academic Editor

PLOS ONE

Reviewers' comments:

Reviewer's Responses to Questions

**Comments to the Author**

1. If the authors have adequately addressed your comments raised in a previous round of review and you feel that this manuscript is now acceptable for publication, you may indicate that here to bypass the “Comments to the Author” section, enter your conflict of interest statement in the “Confidential to Editor” section, and submit your "Accept" recommendation.

Reviewer #1: All comments have been addressed

2. Is the manuscript technically sound, and do the data support the conclusions?

Reviewer #1: Yes

3. Has the statistical analysis been performed appropriately and rigorously? 

Reviewer #1: Yes

4. Have the authors made all data underlying the findings in their manuscript fully available?

Reviewer #1: Yes

5. Is the manuscript presented in an intelligible fashion and written in standard English?

Reviewer #1: Yes

6. Review Comments to the Author

Reviewer #1: This is an excellent contribution. I'll be pleased to see it available for frequent reference in my own research and am certain it will be widely referenced in public policy literature.

7. PLOS authors have the option to publish the peer review history of their article (what does this mean?). If published, this will include your full peer review and any attached files.

Reviewer #1: No

---

## [Editor Report · Acceptance letter]

16 Aug 2022

PONE-D-22-05652R1 

Infectious disease surveillance in U.S. jails: findings from a national survey 

Dear Dr. Maner:

I'm pleased to inform you that your manuscript has been deemed suitable for publication in PLOS ONE. Congratulations! Your manuscript is now with our production department. 

Kind regards, 

on behalf of

Dr. Sungwoo Lim 

Academic Editor

PLOS ONE